# Fine-tuned LLMs Know More, Hallucinate Less
# with Few-Shot Sequence-to-Sequence Semantic Parsing over Wikidata

**Silei Xu*  Shicheng Liu*  Theo Culhane  Elizaveta Pertseva**
**Meng-Hsi Wu[1]  Sina J. Semnani  Monica S. Lam**
Computer Science Department, Stanford University
Stanford, CA
`{silei, shicheng, tculhane, pertseva, sinaj, lam}@cs.stanford.edu`
[1]Ailly.ai
`jwu@ailly.ai`

## Abstract

While large language models (LLMs) can answer many questions correctly, they can also hallucinate and give wrong answers. Wikidata, with its over 12 billion facts, can be used to ground LLMs to improve their factuality.

This paper presents WikiWebQuestions, a high-quality question answering benchmark for Wikidata. Ported over from WebQuestions for Freebase, it consists of real-world data with SPARQL annotation.

This paper presents a few-shot sequence-to-sequence semantic parser for Wikidata. We modify SPARQL to use the unique domain and property names instead of their IDs. We train the parser to use either the results from an entity linker or mentions in the query. We fine-tune LLaMA by adding the few-shot training data to that used to fine-tune Alpaca.

Our experimental results demonstrate the effectiveness of this methodology, establishing a strong baseline of 76% and 65% answer accuracy in the dev and test sets of WikiWebQuestions, respectively. By pairing our semantic parser with GPT-3, we combine verifiable results with qualified GPT-3 guesses to provide useful answers to 96% of the questions in dev. We also show that our method outperforms the state-of-the-art for the QALD-7 Wikidata dataset by 3.6% in F1 score.[1]

## 1  Introduction

Large language models (LLMs) such as GPT-3 can answer open-domain questions without access to external knowledge or any task-specific training examples. However, LLMs are prone to hallucinate (Bang et al., 2023), while using a convincing and confident tone. This may cause significant harm as people increasingly accept LLMs as a knowledge source (Goddard, 2023; Weiser, 2023).

---

*Equal contribution

[1]Code, data, and model are available at https://github.com/stanford-oval/wikidata-emnlp23

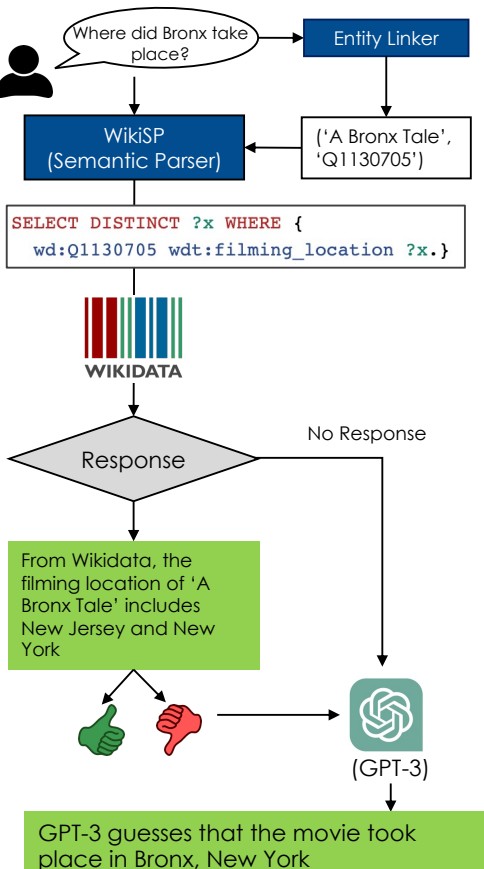

Figure 1: An Overview of WikiSP. An entity linker is used to link entities in the user query to their unique ID in Wikidata; e.g. "A Bronx Tale" is linked to entity ID "Q1130705". The query and entity linker outputs are fed to the WikiSP semantic parser to produce a modified version of SPARQL, where property IDs (e.g. "P915") are replaced by their unique string identifiers (e.g. "filming_location"). If applying the query to Wikidata fails to return a result, we default to GPT-3, labeling the result as a GPT-3 guess. Returned answers are presented in the context of the query, so the user can tell if the answer is acceptable; if not, we also show the guess from GPT-3. Here WikiSP mistakenly uses "filming_location" instead of "narrative_location"; the user detects the mistake, thumbs down the answer, and the GPT-3 answer is provided.

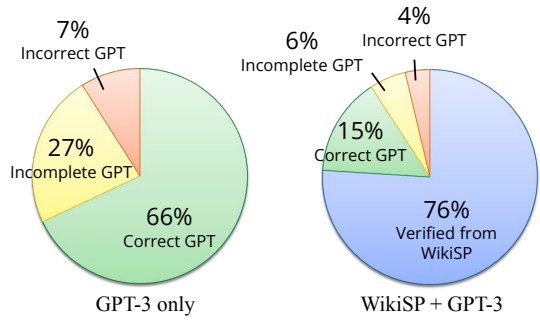

Figure 2: Distribution of correct, incomplete, and incorrect answers for the WikiWebQuestions dev set, when GPT-3 is used alone and when combined with WikiSP.

In contrast, traditional knowledge base question answering (KBQA) is grounded with a given knowledge base. Semantic parsing (SP) has been widely used to tackle this challenging task, where the questions are first parsed into a logical form and then executed to retrieve answers from the knowledge base. It has better interpretability than GPT-3 and other information-retrieval-based approaches (Dong et al., 2015; Miller et al., 2016; Sun et al., 2018, 2019) where answers are predicted directly.

To handle large knowledge bases, previous SP-based approaches tend to use a multi-stage pipeline of sub-tasks, starting with extracting the relevant subgraph based on entities detected in the questions (Yih et al., 2015; Luo et al., 2018). Such an approach struggles with questions that have a large search space and fails to understand questions that refer to information missing in the knowledge graph. Having to retrieve the relevant subgraphs to create the logical form conflates query resolution with semantic parsing, rendering classical query optimization inapplicable.

End-to-end seq2seq translation, on the other hand, has mainly been used on schemas of relatively small relational databases (Yu et al., 2018; Xu et al., 2020a,b) and web APIs (Campagna et al., 2017; Su et al., 2017). To handle large knowledge graphs, recent work proposed retrieving (1) information on linked entities, (2) exemplary logical forms relevant to the query (Gu et al., 2021; Ye et al., 2022), and (3) schemas as context to semantic parsing (Shu et al., 2022). Others use induction or iterative methods to generate complex logical forms (Cao et al., 2022b; Gu and Su, 2022).

## 1.1 Few-Shot Seq2Seq Semantic Parsing

This paper investigates how we can leverage large language models (LLMs) to create seq2seq neural semantic parsers for large knowledge bases such as Wikidata.

Pretrained with the internet corpora, LLMs are already familiar with the syntax of formal query languages such as SQL (Hu et al., 2022; Poesia et al., 2022; Li et al., 2023; An et al., 2023; Nan et al., 2023; Arora et al., 2023). When given simple SQL schemas, they can perform zero-shot semantic parsing of simple natural language queries into formal queries. Unlike Freebase, the KB used in most of the KBQA semantic parsing research, Wikidata does not have a pre-defined schema, making it a much harder problem. It has 150K domains, 3K applicable properties, and 107M entities, each of the properties and entities are uniquely identified with PIDs and QIDs, respectively. While zero-shot LLMs can generate SPARQL queries for the easiest and most common questions, they do not know all the PIDs and QIDs, and nor is it possible to include them in a prompt.

This paper presents WikiSP, a few-shot sequence-to-sequence semantic parser for Wikidata that translates a user query, along with results from an entity linker, directly into SPARQL queries. To handle the 100M+ entities in Wikidata, we train the parser to use either the entity linker results or a mention in the query; to handle the 150K domains and 3K applicable properties, we modify SPARQL to use domain and property names instead of their unique QIDs and PIDs, respectively. We fine-tune a LLaMA (Touvron et al., 2023) with a few-shot training set along with the instructions used to fine-tune Alpaca (Taori et al., 2023).

## 1.2 A New Dataset: WikiWebQuestions

Most of the widely-used high-quality benchmarks for KBQA are based on Freebase (Bollacker et al., 2008) which has been shut down since 2015. With outdated knowledge, it is hard to compare the results with modern LLMs such as GPT-3, since answers have changed over time for most of the questions. Wikidata, despite being the largest and most popular knowledge base nowadays, has very few datasets annotated with SPARQL queries; they are either extremely small (Usbeck et al., 2017) or synthetic (Saha et al., 2018).

We migrated the popular WebQuestionsSP (Yih et al., 2016) benchmark from Freebase to Wikidata, with updated SPARQL and up-to-date answers from the much larger Wikidata.

## 1.3 Complementing Large Language Models

Trained on Wikipedia and all of the internet, LLMs can answer many questions directly. Unfortunately, the user cannot tell if the answers are correct, thus requiring them to fact-check every answer.

Unlike humans, GPT-3 always sounds definitive even when they are wrong by providing specific and plausible facts. For example, on the question "what is the biggest country in Europe by population?", GPT-3 answers "Germany", when the answer is "Russia". Or, on the question, "where does the name Melbourne come from?" GPT-3 answers "Melbourne comes from the Latin word 'melburnum' meaning 'blackburn' or 'blackbird'.", but in reality, Melbourne is named after William Lamb, 2nd Viscount Melbourne. It is not possible to tell when GPT-3's answers are wrong, and every answer needs to be fact-checked.

Semantic parsers can be used to complement LLMs as they are interpretable; their results are grounded in Wikidata, which we assume to be correct. It is possible for semantic parsers to misunderstand a query, but by providing the answer in the context of the query, the user can spot the error.

We propose getting the best of both worlds by answering the question with WikiSP if possible. Otherwise, we report GPT-3's guesses by prefacing it with: "GPT-3 guesses that" (Figure 1). In this way, the user can have full confidence with the answers from the former, while also benefiting from the latter. It is easier for users to fact-check an answer than trying to find the answer.

## 1.4 Contributions

**WikiWebQuestions, a high-quality semantic parsing dataset for Wikidata**, migrated from the popular WebQuestions dataset for Freebase.

**WikiSP, a few-shot Seq2Seq semantic parser** by fine-tuning LLaMA with a few shot training set. We improve the learnability of SPARQL queries by replacing the IDs of properties and domains with their unique names; we tolerate errors in entity linking by accepting mentions in the queries as entities. We establish a first, strong baseline of 76% and 65% answer accuracy for the dev set and test set of our new WikiWebQuestions benchmark, respectively. We also demonstrate that our method surpasses the state of the art for QALD-7 wikidata set by 3.6% in F1 score.

We **improve GPT-3's trustworthiness** by first returning interpretable results from semantic parser

and backing it up with GPT-3 guesses. WikiSP can provide verifiable results for WikiWebQuestions 76% of the time and improves the guesses by GPT-3, resulting in errors only 4% of the time (Figure 2).

## 2 Related Work

### 2.1 KBQA

The KBQA task aims to make large knowledge bases accessible by natural language. One common approach is semantic parsing where a natural language query is translated into a formal logical form, which is then executed to retrieve an answer from the knowledge base. To handle large KBs, one method is to formulate SP as a multi-staged search problem by retrieving entities and expanding the graphs according to the relationships between their properties and the query (Yih et al., 2015, 2016; Luo et al., 2018). Lan and Jiang (2020) add constraints to the staged query graph generation method. Another popular method is to use seq2seq models obtained by fine-tuning pretrained language models. Das et al. (2021) first find other queries that contain semantically similar subparts, and construct a new logical form by combining the similar subparts of the found queries. Ye et al. (2022) search over the KB based on predefined rules to derive a set of candidate logical forms, rank them, and generate the final logical form. Cao et al. (2022b) first generate a "sketch" program and then fill in its arguments. Gu and Su (2022) use dynamic program induction to generate query structures. Based on a user query, Shu et al. (2022) retrieve entities, example logical forms, and related schema. Unlike FreeBase, Wikidata does not have a fixed schema.

Another approach to KBQA is based on graph retrieval (Dong et al., 2015; Miller et al., 2016; Sun et al., 2018, 2019; Mavromatis and Karypis, 2022; Sen et al., 2021; Vivona and Hassani, 2019; Verga et al., 2021). It predicts the answers directly within the subgraph extracted based on the topic entity in the question. Yu et al. (2023) combine semantic parsing with retrieval and achieve the state-of-the-art on the WebQuestionsSP dataset (Yih et al., 2016). However, retrieval-based methods cannot handle entire categories of questions, such as questions with no available answer and questions like "the tallest mountain" where no entities are mentioned by name. They have poor interpretability and do not support query optimization.

## 2.2 KBQA Benchmarks

Most of the early KBQA benchmarks are based on Freebase (Berant et al., 2013; Yih et al., 2016; Talmor and Berant, 2018). Recently, new benchmarks have been created for Wikidata (Cao et al., 2022a; Saha et al., 2019). However, these benchmarks are created using rule-based synthesis or paraphrases, which are easier for semantic parsers. CSQA collects human-written questions for single triples and constructs complex questions using fixed rules with very limited natural language variety (Saha et al., 2019). KQA Pro first synthesizes queries with canonical natural language and then crowdsources human paraphrases (Cao et al., 2022a). Campagna et al. (2019) show that a model can achieve significantly higher accuracy over paraphrased data compared to real-world data even for untrained queries. Thus, we base our WikiWebQuestions dataset on WebQuestionsSP (Yih et al., 2016), where data are collected from real-world users using the Google Suggest API.

## 2.3 LLMs for Semantic Parsing

Shin et al. (2021) show the promise of few-shot prompting LLMs for semantic parsing. They use constrained decoding to enforce the syntax of the formal language, and achieve comparable results with a smaller fine-tuned BART model (Lewis et al., 2020) on datasets with small database schemas. Rubin et al. (2022) fine-tune a small retriever to obtain the most relevant few-shot examples to use for each input. Niu et al. (2023) use a few-shot prompted Codex model to break down the natural language input to make the task easier for a smaller semantic parser. LLMs have also been applied to semantic parsing on relational databases (Hu et al., 2022; Poesia et al., 2022; Li et al., 2023; An et al., 2023; Nan et al., 2023; Arora et al., 2023). The schemas used in these projects are very small when compared to Wikidata.

## 2.4 Entity Linking

Entity linking involves finding the named entities in a query, and linking them to the corresponding entities in the knowledge graph so that the query can be executed using the proper entities as reference points. The current state-of-the-art entity linking model on the WebQuestionsSP dataset is ReFinED (Ayoola et al., 2022). They use a bidirectional transformer on the query to predict the most likely mentions of named entities within a query, and then combine that information with embeddings computed over every entity in the knowledge base to predict which entity the mention is most likely to be referring to. Prior to ReFinED, the state-of-the-art was ELQ (Li et al., 2020). They similarly generate embeddings for each entity in the knowledge base, and then use the predicted mentions of entities combined with these predicted embeddings to generate likely entities.

## 3 Semantic Parsing for Wikidata

Wikidata is the largest public knowledge base with over 12 billion facts represented by subject-predicate-object triples using 100+ million entities and 10,000 properties. 3,000 of the properties are useful for answering natural language questions, whereas the rest are used to link data in Wikidata with external library catalogs and database IDs.

Entities and properties are given unique identifiers, QIDs and PIDs, respectively. For example, the fact that Joe Biden is the president of the US can be represented as a triple (Q6279, P39, Q11696), where P39 is the PID for property *position held*, Q6279 and Q11696 are QIDs for Joe Biden and the president of the United States, respectively.

### 3.1 Query Format

Unlike relational databases and Freebase, Wikidata has no predefined domains or types. Any entity can have an arbitrary set of properties. However, even though Wikidata is property-based, all named entities have one or more *instance of* properties to some *domain* entity; domain entities are organized into a hierarchy with the *subclass of* property.

Note that the names of domain entities and properties are unique. Non-domain entities, on the other hand, can be ambiguous. For example, "Lincoln" can refer to the president, a car brand, a sparrow, an aircraft, and many different cities.

We posit that it is impossible for LLMs to memorize the QIDs and PIDs for domains and properties. We modify the format of SPARQL queries to use the more mnemonic property name, instead of its PID. Similarly, we use entity names for domains. For example, the original SPARQL for the query "What car models does GM make?" is

```
SELECT DISTINCT ?x WHERE {
?x wdt:P31/wdt:P279* wd:Q3231690.
?x wdt:P176 wd:Q81965. }
```

This says that we are seeking $x$, where $x$ is transitively either an *instance of* (wdt:P31)

or a *subclass of* (wdt:P279) of an automobile model (wd:Q3231690), and $x$ has General Motors (wd:Q81965) as the *manufacturer* (wdt:P176). Note wdt is the prefix for Wikidata property, and wd is for Wikidata entity.

With our modification, the query becomes:

```
SELECT DISTINCT ?x WHERE {
?x wdt:instance_of/wdt:subclass_of*
wd:automobile_model.
?x wdt:manufacturer wd:Q81965. }
```

For non-domain entity QIDs, we also accept a string in lieu of a QID in case of entity linking errors. At inference time, we use simple heuristics to resolve the string to a QID before applying the query. For example, "wd:Q81965" in the query may be replaced with "wd:GM". See Section 3.2.2 for more details.

Normally, we refrain from changing standard query notations since LLMs have been pretrained on them. However, we posit that learning this new syntax is much easier than learning the PIDs and QIDs. Our experimentation with few-shot prompting suggests that LLMs can easily adjust to this format.

## 3.2 Entity Linking

Linking entities for WikiWebQuestions is particularly difficult. First, since the dataset is collected from real-world questions without prompting the users for more information, users tend to refer to their entities of interest without using their full names. Second, the questions are generally short with very limited context, making it harder to disambiguate among entities with similar names. Lastly, many QIDs in Wikidata are used to represent terms not generally known as "named entities". For example, domain entities are often ignored by entity linker models, as in "What is the biggest country in Europe by population?", both "country" (Q6256) and "Europe" (Q46) are required to construct the correct SPARQL, but entity linkers only provide "Europe" and ignore "country".

### 3.2.1 Semantic Parsing with Entity Linking

To handle ambiguous entities, we use an entity linker to first find the domain names and QIDs of the entities mentioned in the text. We train a semantic parser that accepts users' input along with the results produced by the entity linker.

Formally, given a user input $T$, and a set of entity linker results $\langle e, q \rangle$, where $e$ is the name (default label) Wikidata gives to an entity and $q$ is its QID,

the semantic parser produces the semantic parse of $T$ in our modified SPARQL format.

For the example above, the SOTA ReFinED entity linker (Ayoola et al., 2022) returns $\{\langle$General Motors, Q81965$\rangle\}$. Unfortunately, it misses the entity automobile model (Q3231690), a term not usually considered to be an entity.

### 3.2.2 Recovering from Entity Linker Errors

We want our semantic parser to be able to recover from mistakes by an entity linker. That is, the semantic parser should use entity linking when it is helpful, but it should still try to predict the right logical form when the linker fails.

The semantic parser is trained to accept, along with the user query, an *optional* set of *potentially* useful QIDs from the entity linker. We include samples where some of the supplied linked entities are not used in the gold answer, as well as samples where there are missing linked entities. For the latter, we use mentions in the original query in lieu of the QIDs. At inference time, we use the mentions to look up the QIDs in Wikidata. If multiple matches exist, the most popular entity is returned. An example is shown in Appendix A.

With the above example where the entity linker misses "automobile model", the semantic parser is likely to predict "car model" by copying from the user query. We search "automobile model" among aliases in domains to find the correct QID. This design allows the model to potentially recover from entity-linking failures.

## 4 WikiWebQuestions (WWQ) Dataset

Despite being the most popular large knowledge base for a long time, existing benchmarks on Wikidata with labeled SPARQL queries are unfortunately either small or of low quality. On the other hand, benchmarks over the deprecated Freebase still dominate the KBQA research with better-quality data. For example, the WebQuestions (Yih et al., 2015) dataset was collected by using Google Search API instead of human paraphrasing or synthesis. As a result, it is much more natural and truly reflects the real-world questions users may ask. This dataset is later annotated with SPARQL over Freebase, named WebQuestionsSP (Yih et al., 2016). Examples with no legitimate SPARQL to retrieve answers from Freebase are dropped. In total, WebQuestionsSP consists of 3098 examples in the training set and 1639 in the test set.

We migrated WebQuestionsSP, the best collection of natural language questions over a general knowledge graph, from Freebase to Wikidata, with the help of an automatic tool we developed, based on Google's entity mapping[2] and Wikidata's relation mapping[3]. About 60% of the dataset was automatically converted. One of the authors of this paper, who did not participate in model tuning, manually converted those instances that failed to convert automatically.

## 4.1 Migrating WebQuestionsSP to Wikidata

Here are the major decisions we made in migrating WebQuestionsSP dataset to Wikidata. While much bigger, Wikidata does not necessarily contain all the information available in Freebase. For example, it lacks countries' trade partners, hence we drop all such questions from the WebQuestionsSP dataset.

If multiple paths can lead to the correct answer, we choose the path that provides the most complete answers and has the best availability among entities in the same domain. For example, when asking for books written by an author X, we can either search for books whose *author* is X or find *notable works* of X that are books. While the latter is more efficient, the property *notable works* is not always available for all authors and it often does not provide a complete list. Thus, we annotate such examples using the former representation.

We also cleaned up the original dataset. The dataset contained questions like "who does Ronaldinho play for now in 2011?". We drop the appended year as it conflicts with "now" in the utterance, and it would refer to the live information in Wikidata.

In total, we dropped 9% of the examples from WebQuestionsSP and created a training, dev, and test set of 2431, 454, and 1431 samples, respectively. Given that Wikidata has 100 million entities and 3,000 useful properties for answering questions, the training data set is woefully inadequate and can be considered as a "fewshot" training set at best.

## 5 Implementation

This section discusses the implementation details of the entity linker and the WikiSP semantic parser.

## 5.1 Entity Linking

We use ReFinED (Ayoola et al., 2022) for entity linking, which is the current state of the art for WebQuestionsSP. As discussed before, Wikidata treats many common terms such as "country" as named entities and assigns them QIDs. To fine-tune ReFinED to learn such terms, we add the question and entity pairs from the training set of WikiWebQuestions to the data used to train ReFinED's questions model.

We run 10 epochs of finetuning using the default hyperparameters suggested by Ayoola et al. (2022). For each identified entity, we provide the mention in the original utterance, the QID, as well as its domain in plain text. The information is appended to the utterance before being fed into the neural semantic parsing model.

## 5.2 The WikiSP Semantic Parser

We prepare the training data with entities provided by fine-tuned ReFinED. Comparing with the gold entities, ReFinED provides extra entities in 215 cases, while missing at least one entity in 137 cases. When ReFinED failed to produce the correct entities, we replace the missing QIDs in the logical form with the corresponding mention of the entity in the question. During evaluation, if a mention of an entity is predicted by the model, we look up the QID using the Wikidata "wbsearchentities" API [4].

We fine-tune LLaMA with 7B parameters because it has been shown to perform well despite its relatively small size (Touvron et al., 2023). We include the Alpaca (Taori et al., 2023) instruction following data, which was derived using the self-instruct (Wang et al., 2023) method, in our training data. The training data samples in WikiWebQuestion start with the following instruction: "Given a Wikidata query with resolved entities, generate the corresponding SPARQL. Use property names instead of PIDs.". We concatenate the resolved entities and the user utterance together as input. We up-sample the WikiWebQuestion fewshot set 5 times and train for 3 epochs using 2e-5 learning rate and 0.03 warmup ratio.

## 5.3 Executing Queries on Wikidata

SPARQL queries are used to retrieve answers from the Wikidata SPARQL endpoint[5]. Since Wikidata

---

[2]https://developers.google.com/freebase
[3]https://www.wikidata.org/wiki/Wikidata:WikiProject_Freebase/Mapping
[4]https://www.wikidata.org/w/api.php?action=wbsearchentities
[5]https://www.wikidata.org/wiki/Wikidata:SPARQL_query_service

|                | EM   | F1   |
|----------------|------|------|
| WikiSP (ours)  | 65.5 | 71.9 |

Table 1: Results of WikiSP on the WWQ test set.

is actively being updated, the gold SPARQL can be easily re-executed to acquire up-to-date answers, allowing the benchmark to compare with forthcoming iterations of large language models.

## 6 Experiments

In this section, we evaluate WikiSP on WikiWebQuestions and demonstrate how it can be used to complement large language models such as GPT-3.

### 6.1 Semantic Parser Results

We evaluate our model with two different answer accuracy metrics: (1) exact match (EM): the percentage of examples where the answers of the predicted SPARQL exactly match the gold answers, and (2) Macro F1 score (F1): the average F1 score for answers of each example. The evaluation results are shown in Table 1. Our approach achieves a 65.5% exact match accuracy and a 71.9% F1 score on the WWQ dataset.

As a reference, the current state-of-the-art result on the original WebQuestionsSP dataset for Freebase is 78.8% F1 (Yu et al., 2023). The result was obtained with a combination of semantic parsing and retrieval. The WikiWebQuestions dataset is slightly different, as discussed above. More significantly, unlike Freebase, Wikidata does not have a fixed schema and ours is an end-to-end, seq2seq semantic parser.

### 6.2 Ablation Experiments

#### 6.2.1 Entity Linking

Our first ablation study evaluates the need for entity linking with ReFinED, by replacing it with simply using the LLM to detect entities as mentions. In this experiment, all entity IDs in the training data are replaced by their mentions; during inference, we map the predicted entities to their actual QIDs according to Section 3.2.2.

The results show that replacing the neural entity linker with just using mentions reduces the exact match by 9.1% and the F1 score by 9.3%. This suggests that entity linking is important.

#### 6.2.2 Allowing Mentions as Entities

Our logical form is designed to recover from entity linking errors by allowing entities be specified by

|                                              | EM   | F1   |
|----------------------------------------------|------|------|
| WikiSP (ours)                                | **75.6** | **76.9** |
| No Entity Linking                            | 66.5 | 67.6 |
| No mentions, trained with ReFinED            | 73.3 | 75.0 |
| No mentions, trained with Oracle entities    | 72.2 | 73.4 |
| PIDs and QIDs for properties & domains       | 73.6 | 74.7 |

Table 2: Ablation results of WikiSP on the WWQ dev set.

a mention, as an alternative to a QID. Our ablation study on this feature tested two training strategies:

**ReFinED.** The entity linker tuples are produced by fine-tuned ReFinED, which may be missing entities in the gold target. The data show that generating unseen QIDs is needed for missing entities.

**Oracle.** The entity linker tuples are exactly all the entities used in the gold. The model would only encounter missing QIDs at test time when ReFinED fails to generate all the necessary QIDs.

The answer accuracy of the model using entity linked tuples from **ReFinED** ("No mentions, trained with ReFinED" in Table 2) lags by 2.3% when compared against our best model. The model using **Oracle** ("No mentions, trained with Oracle entities" in Table 2) lags by 3.4%. These results indicate that allowing mentions is useful for recovering from entity linking errors.

#### 6.2.3 Names vs. IDs for Properties & Domains

Our logical form replaces PIDs with property names, and domain-entity QIDs with the domain names. Here we evaluate the effectiveness of this query format. We compare our approach with the original SPARQL where all properties and entities are represented with PIDs and QIDs. Our ablation study shows that our representation with property names and domain names improves the answer accuracy by 2.0% (Table 2). This shows that LLMs can adapt to changes in query notation with fine-tuning, and it is easier to learn names than remembering random IDs. If we did not allow mentions in the predicted logical form, the replacement of QIDs with their names is likely to be more significant.

### 6.3 Complementing GPT-3

LLMs like GPT-3 can answer many questions on general knowledge correctly; however, they may also hallucinate. WWQ is representative of popular questions, so we expect GPT-3 to perform well. We use text-davinci-002 with the temperature set to 0 to evaluate GPT-3's performance on WWQ.

On the dev set of WWQ, GPT-3 answers 66.4%

of the questions correctly and provides incomplete answers to 26.5% of the questions. For example, when asked "What does Obama have a degree in?", GPT-3 correctly identifies President Obama's political science degree, but fails to mention his law degree. In total, GPT-3 gives wrong answers to 7.1% of the questions.

For this dev set, we can give definitive answers to 75.6% of the questions with WikiSP (Table 2). For the rest of the questions (24.4%), accounting for the overlap between the GPT-3 and our semantic parser's results, the percentages of guessing correctly, incompletely, and incorrectly are at 15.2%, 5.5%, and 3.7%, respectively (Figure 2).

In summary, the combination of GPT-3 and WikiSP makes it possible to give a definitive, correct, and complete answer three quarters of the time for the dev set. Users can also benefit from GPT-3's guesses the rest of the time at a 3.7% error rate, which is about half of the original error rate.

## 6.4   Error Analysis

We analyzed the 111 examples in the WWQ dev set where the model failed.

### 6.4.1   Acceptable Alternative Results (18.0%)

Our analysis shows that 18.0% of the "errors" can actually be deemed to be correct.

**Reasonable alternate answers (11.7%).** In 11.7% of the cases, the model predicts an alternative interpretation to the question and returns a reasonable answer that is different from the gold. For example, the gold for question "what did Boudicca do?" uses the *position held* property, while the model predicts *occupation* property. Both are considered valid answers to the question.

**Reasonable alternative SPARQL but no answer was retrieved (6.3%).** In another 6.3% of cases, the model predicts a reasonable alternative SPARQL, but the SPARQL returns no answer. Sometimes, since the information for the "correct" property is missing, the question is represented with a similar property. For example, since *residence* property is missing for Patrick Henry, the gold SPARQL for "where did Patrick Henry live?" uses *place of birth* instead, while our model predicts *residence*.

### 6.4.2   Errors in Entity Linking (35.1%)

The biggest source of errors is entity linking. Entity linker failed to provide the correct entities in 35.1% of the failed examples. While WikiSP can potentially recover from missing entities, it cannot recover from incorrect entities. This is especially common for character roles, as some character roles have different entities for books and movies or even different series of movies. Sometimes WikiSP located the correct mention from the question, but the lookup failed. For example, the model located the mention of the event "allied invasion of France" in question "where did the allied invasion of France take place?", but failed to find the corresponding entity from Wikidata by the name.

### 6.4.3   Errors Beyond Entity Linking

Semantic parsing in Wikidata is challenging as there are no predefined schemas, and there are 150K domains and 3K applicable properties. Some representative mistakes include the following:

**Wrong property (17.1%).** 17.1% of the errors are caused by predicting the wrong property. Some of the examples require background knowledge to parse. For example the answer of the question "what did martin luther king jr do in his life?" should return the value of *movement*, while the model predicts *occupation*. Properties are a challenge in Wikidata because as illustrated here which property to predict depends on the entity itself.

**Missing domain constraint (5.4%).** Another common problem is missing the domain constraint. For example, the model correctly identifies that property *shares border with* should be used for question "what countries are around Egypt?". However, it does not limit the answer to countries only, thus extra entities are returned.

## 7   Experiment with QALD-7

For another evaluation of WikiSP, we apply our model on Task 4 from QALD-7 (Usbeck et al., 2017) dataset. QALD-7 is part of the QALD (Question Answering over Linked Data) which is a series of challenges started in 2011 known for their complex, manually created questions. It mainly focuses on DBpedia, but QALD-7's Task 4 is engineered for Wikidata. The task includes 100 train examples, which we use to fine-tune our model and 50 test examples. There is no dev set.

We choose QALD-7 as it is a manually crafted dataset with complex questions. We avoid datasets built on synthetic or human-paraphrased data, such as CSQA (Saha et al., 2018) and KQA-Pro (Cao et al., 2022a). As they have limited natural language variety between the training and evaluation

| | EM | F1 |
|---|---|---|
| STAGG (Yih et al., 2016) | - | 19.0 |
| GGNN (Sorokin and Gurevych, 2018) | - | 21.3 |
| WDAqua (Diefenbach et al., 2017) | - | 40.0 |
| WikiSP (Ours) | **38.0** | **43.6** |

Table 3: Evaluation results of WikiSP on QALD-7 Task 4 and comparison with prior work.

data, models can get artificially high accuracy. For example, a simple BART based model can achieve over 90% accuracy on KQA-Pro even without an entity linking module (Cao et al., 2022a).

The QALD-7 test set provides both the SPARQL queries as well as the answers. To double-check the correctness of the QALD-7 dataset, we applied the 50 gold queries of the test set to Wikidata and found that 4 did not return an answer. We hypothesize that the discrepancy is caused by the change in Wikidata structure/quantity of information. We evaluate WikiSP by comparing the answers where possible, and by comparing the generated SPARQL syntactically otherwise.

For this experiment, we use the same hyper-parameters and data format as described in Section 5.3. In addition to the training data for WikiSP, we also include the QALD-7 train samples, upsampled 20 times.

## 7.1 QALD-7 Results

Our model achieves 38% accuracy on the QALD-7 dataset and outperforms the F1 score of the state-of-the-art WDAqua (Diefenbach et al., 2017) by 3.6%, as shown in Table 3. Note that WDAqua is based on retrieval, whereas WikiSP is based on sequence-to-sequence semantic parsing. QALD-7 (Usbeck et al., 2017) reports WDAqua as the winner of the leaderboard with 55.2 F1, however the authors of WDAqua reported 40.0 F1 in their papers (Diefenbach et al., 2017).

## 7.2 Complementing GPT-3 on QALD-7

Similar to WWQ, we also assess the combination of GPT with WikiSP on QALD-7 as shown in Figure 3. The GPT model used was "text-davinci-002". Since there is no validation set and the test set is already very small, one of the authors who was not involved in training or finetuning the model evaluated GPT-3 on the test set.

GPT-3 is fully accurate on 62% of the questions, 20% incomplete, and 18% wrong. With our approach, we can provide 38% verifiably good answers from WikiSP; the guesses of GPT-3 get an

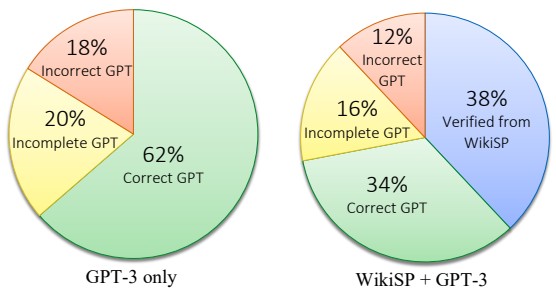

Figure 3: Distribution of correct, incomplete, and incorrect answers for the QALD-7 test set, when GPT-3 is used alone and when combined with WikiSP.

additional 34% correct, 16% incomplete, and only 12% wrong.

## 7.3 Discussion

We did not conduct error analysis on the performance of QALD-7 as it has no dev set. The author evaluating GPT-3 noted that the test set of QALD-7 is much more complicated than the training data (of just 100 samples), with most of the queries containing multiple properties. This explains the lower accuracy of WikiSP on QALD-7 when compared to WikiWebQuestions, which has a few-shot training data set with a similar distribution as the test set.

This result suggests that the performance of WikiSP depends heavily on a good few-shot training data for fine-tuning the LLMs. We hypothesize that we can increase the performance of WikiSP in handling less popular questions with a better, possibly synthesized, training dataset.

## 8 Conclusion

We have created a new high-quality benchmark, WikiWebQuestions, for large knowledge-base question answering. The dataset is based on the popular WebQuestionsSP dataset with natural questions, annotated with SPARQL for Wikidata.

We establish a first, strong baseline of 65% answer accuracy and 72% F1 score for WikiWebQuestions. This is achieved by fine-tuning LLaMA with a few-shot training data set using a SPARQL query format modified for semantic parsing.

We show that we can reduce the hallucination of large language models like GPT-3 by grounding it with a semantic parser. For the dev set of WikiWebQuestions, this combination approach provides useful information for 96% of the questions in the dev set of the benchmark. More importantly, it generates verifiable answers for 76% of the questions.

## Limitations

While applications of large language models seem to expand every day, this paper mainly focuses on factoid question answering. Long-form text generation, for example, is outside the scope of the experiments of this paper, but the methodology described here may be extended to this setting in the future. Even though knowledge bases are an important source of facts, a large portion of the knowledge available in digital form (e.g. Wikipedia, news articles, etc.), is not organized into knowledge bases. As such, the results of this paper can be considered complementary to the larger body of fact-checking research based on free text.

Our semantic parser can be used to verify answers from LLMs. However, this additional round of running the semantic parser and querying Wikidata increase the response latency, which may be noticeable by end-users of such systems.

All of our datasets and experiments are conducted for English. Expanding to other languages, while possible (Moradshahi et al., 2020) are outside the scope of this work.

Our experiments were performed using GPT-3 (davinci-002) as that was what we had access to when we started the project. Undoubtedly, the later LLMs will produce better results. Nonetheless, the need to have verifiable results based on live database accesses will remain.

## Ethical Considerations

LLMs are used by millions of people everyday. We hope that this line of work will help make them more reliable for everyone, mitigating some of their potential downsides, and giving users access to more accurate information. Our use of Wikidata will enable future researchers and developers to connect their systems with a large, diverse and live knowledge graph that is updated every day. We do not anticipate any harm resulting from the methods introduced in this work.

We did not crowdsource any datasets for this paper, as the questions are converted from a previous dataset and all the re-annotation and analysis is done by the authors.

To conduct experiments in this paper, we used an estimated total of 60 NC96ads-A100 GPU hours on Microsoft Azure. Each finetuning experiment takes roughly 3 hours, and we conducted roughly 20 experiments to arrive at the results in this paper.

## Acknowledgements

This work is supported in part by the National Science Foundation, the Alfred P. Sloan Foundation, the Verdant Foundation, Microsoft Azure AI credit, KDDI, JPMorgan Chase, and the Stanford Human-Centered Artificial Intelligence (HAI) Institute. We also thank the reviewers for their valuable comments and suggestions.

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

# A Examples of Recovering from Entity Linking Errors

Here, we illustrate our proposal of using entity mentions to recover from entity linking errors. In the training set, we have the following example:

- Query: What year did giants win the world series?
- Original Gold SPARQL:

  SELECT DISTINCT ?x WHERE {
  ?y wdt:sports_season_of_league_or_competition
  wd:Q265538;
  wdt:winner wd:Q308966;
  wdt:point_in_time ?x. }

- Gold Entity linker result:

  World Series (QID Q265538),
  San Francisco Giants (QID Q308966);

- ReFinED result:

  San Francisco Giants (QID Q308966);

Here, the ReFinED entity linker model fails to identify the "World Series" entity. Our proposal of mentions gives the semantic parser a chance to recover from entity linker failures. To train the parser to generate mentions, our training includes samples like this:

- Query: what year did giants win the world series?
- ReFinED result:

  San Francisco Giants (QID Q308966);

- Gold target:

  SELECT DISTINCT ?x WHERE {
  ?y wdt:sports_season_of_league_or_competition;
  wd:world_series;
  wdt:winner wd:Q308966;
  wdt:point_in_time ?x. }

The gold query mentions "world_series". At inference time, our heuristics use the predicted mention to look up the actual Wikidata entity. For example, if wd:world_series is predicted at inference time, our heuristics maps it back to wd:Q265538.