# OpenReview forum: "Fine-tuned LLMs Know More, Hallucinate Less with Few-Shot Sequence-to-Sequence Semantic Parsing over Wikidata"
_EMNLP/2023/Conference — EMNLP 2023 Main_

### Official Review · Reviewer_2RKA · 2023-08-04

**Soundness:** 3

**Excitement:**

3: Ambivalent: It has merits (e.g., it reports state-of-the-art results, the idea is nice), but there are key weaknesses (e.g., it describes incremental work), and it can significantly benefit from another round of revision. However, I won't object to accepting it if my co-reviewers champion it.

**Paper Topic And Main Contributions:**

This paper is about improving the factuality of large language models (LLMs) by grounding them in Wikidata.

The main contributions of the paper are:

1. The creation of the WikiWeb-Questions dataset, which serves as a benchmark for large knowledge-base question answering.
2. A simple sequence-to-sequence semantic parser over Wikidata.
3. The pairing of the semantic parser with GPT-3, which provides a combination of verifiable results and qualified guesses that offer useful answers to 96% of the questions in the dev set of the benchmark.

Overall, the paper demonstrates that grounding LLMs with a semantic parser for Wikidata can reduce hallucination and improve the factuality of their answers.

**Reasons To Accept:**

The strengths of this paper include:

1. Introduction of a new dataset, WikiWeb-Questions, which is a high-quality knowledge base question answering benchmark for Wikidata.

2. The paper presents the first effective few-shot sequence-to-sequence semantic parser for Wikidata.

3. The proposed methodology reduces the hallucination of large language models like GPT-3 by grounding it with a semantic parser for Wikidata.

**Reasons To Reject:**

- This paper only conducts experiments on a single dataset (self-collected), but there are many existing KBQA datasets (e.g., WebQuestions, GrailQA) in the literature. Conducting experiments on popular KBQA datasets can make it better understand the effectiveness of the proposed method.

- This paper does not compare with other KBQA methods in the literature.

**Reproducibility:**

4: Could mostly reproduce the results, but there may be some variation because of sample variance or minor variations in their interpretation of the protocol or method.

**Reviewer Confidence:**

3: Pretty sure, but there's a chance I missed something. Although I have a good feel for this area in general, I did not carefully check the paper's details, e.g., the math, experimental design, or novelty.

---

> ### Author Rebuttal · Authors · 2023-08-29
>
> Thank you for your review and valuable comments.
>
> > This paper only conducts experiments on a single dataset (self-collected), but there are many existing KBQA datasets (e.g., WebQuestions, GrailQA) in the literature. Conducting experiments on popular KBQA datasets can make it better understand the effectiveness of the proposed method.
>
> > This paper does not compare with other KBQA methods in the literature.
>
> We focus on question answering on Wikidata and we avoid datasets built on synthetic data or human paraphrase data, such as CSQA ([Saha et al. 2018]( https://ojs.aaai.org/index.php/AAAI/article/view/11332 ))  and KQA-Pro ([Cao et al. 2022]( https://aclanthology.org/2022.acl-long.422/ )). For synthetic or human paraphrase data, due to the extremely limited natural language variety shared between the training and evaluation data, models can get artificially high accuracy. For example, a simple BART model can achieve over 90% accuracy on KQA-Pro even without an entity linking module.
>
> To better evaluate the effectiveness of our method as proposed by the reviews we also evaluated our model on QALD-7 ([Usbeck et al. 2017]( https://svn.aksw.org/papers/2017/ESWC_2017_QALD/public.pdf )). This dataset is part of the QALD (Question Answering over Linked Data) challenges, a popular series of challenges started in 2011 known for their complex, manually created questions. It mainly focuses on DBpedia, but QALD-7 's Task 4 is engineered for Wikidata. The task includes 100 train examples, which we use to finetune our model, and 50 test examples for evaluations.
>
> The QALD-7 test set contains both SPARQL queries as well as the answers. To double-check the correctness of the QALD-7 dataset, we applied the 50 gold queries of the test set to Wikidata and found that 4 did not return an answer. We hypothesize that the discrepancy is caused by the change in Wikidata structure/quantity of information. Thus the results below are reported only on the 46 remaining. Note that the prior work we are comparing to evaluates on all 50.
>
> For this experiment, we use the same hyperparameters and data format as described in Section 5.2. In addition to the training data for WikiSP, we also include the QALD-7 train examples, upsampled by 20 times.
>
> QALD-7 ([Usbeck et al. 2017]( https://svn.aksw.org/papers/2017/ESWC_2017_QALD/public.pdf )) reports that the leader WDAqua gets an F1 score of 0.552, but the paper of WDAqua ([Diefenbach et al. 2017]( https://link.springer.com/chapter/10.1007/978-3-319-69146-6_8 ))  reports 0.40. The runner-up reports an F1 score of 0.427. Our model achieves an F1 score of 0.47. Note that WDAqua is based on retrieval, whereas WIkiSP is based on semantic parsing. No comparable results were reported on semantic parsing. (The differences between retrieval and semantic parsing are discussed in Section 2.1)
>
> One of the authors who was not involved in training or fine-tuning the model evaluated GPT-3 on the test set. GPT-3 is fully accurate on 63% of the questions, 20% incomplete, 16% wrong. WikiSP gets 41% of the questions correct.  So we can provide 41% verifiably good answers; the guesses of GPT-3 get an additional 33% correct, 15% incomplete, and 11% wrong. It was noted that the test set is significantly more complicated than the training data (of just 100 samples), with most of the queries containing multiple properties, thus explaining the lower accuracy. (We do not report a detailed error analysis because there is no dev set).
>
> ### References
>
> [1] Amrita Saha, Vardaan Pahuja, Mitesh Khapra, Karthik Sankaranarayanan, and Sarath Chandar. 2018. [Complex Sequential Question Answering: Towards Learning to Converse Over Linked Question Answer Pairs with a Knowledge Graph](https://ojs.aaai.org/index.php/AAAI/article/view/11332). *Proceedings of the AAAI Conference on Artificial Intelligence, 2018.*
>
> [2] Shulin Cao, Jiaxin Shi, Liangming Pan, Lunyiu Nie, Yutong Xiang, Lei Hou, Juanzi Li, Bin He, and Hanwang Zhang. 2022. [KQA Pro: A Dataset with Explicit Compositional Programs for Complex Question Answering over Knowledge Base](https://aclanthology.org/2022.acl-long.422/). *Proceedings of the 60th Annual Meeting of the Association for Computational Linguistics (Volume 1: Long Papers).*
>
> [3] Ricardo Usbeck, Axel-Cyrille Ngonga Ngomo, Bastian Haarmann, Anastasia Krithara, Michael Röder, and Giulio Napolitano. 2017. [7th Open Challenge on Question Answering over LinkedData (QALD-7)](https://svn.aksw.org/papers/2017/ESWC_2017_QALD/public.pdf).
>
> [4] Dennis Diefenbach, Kamal Singh, and Pierre Maret. 2017. [WDAqua-core0: A Question Answering Component for the Research Community](https://link.springer.com/chapter/10.1007/978-3-319-69146-6_8). *SemWebEval 2017. Communications in Computer and Information Science, vol 769.*

---

### Official Review · Reviewer_CHSV · 2023-08-04

**Paper Topic And Main Contributions:** 1. The paper introduces a novel seman…
**Typos Grammar Style And Presentation Improvements:** The paper reads well.
**Soundness:** 4

**Excitement:**

4: Strong: This paper deepens the understanding of some phenomenon or lowers the barriers to an existing research direction.

**Missing References:**

1. Can you cite sources for "This may cause significant harm as people increasingly accept LLMs as a knowledge source." (line 38-39)

**Questions For The Authors:**

1. Are constrained decoding techniques like PICARD (Scholak2021, https://github.com/ServiceNow/picard) considered for decoding the SPARQL queries?

2. I am not sure how the backoff to GPT3 system would work? When is model not confident enough to backoff to GPT3?

**Reasons To Accept:**

1. WWQ dataset is a useful resource to the community facilitating integration of large KBs with LLMs.

2. The proposed idea of using string representations instead of Wikidata IDs is well-founded and empirically justified.

**Reasons To Reject:**

1. WebQuestionsSP was proposed in 2016 and the answers for that also would have changed (line 104-107)? How does the proposed dataset ensure that the answers are still relevant/correct.

**Reproducibility:**

5: Could easily reproduce the results.

**Reviewer Confidence:**

4: Quite sure. I tried to check the important points carefully. It's unlikely, though conceivable, that I missed something that should affect my ratings.

---

> ### Author Rebuttal · Authors · 2023-08-29
>
> Thank you for your review and valuable comments.
>
> > WebQuestionsSP was proposed in 2016 and the answers for that also would have changed (line 104-107)? How does the proposed dataset ensure that the answers are still relevant/correct.
>
> Thank you for this question. We conducted a thorough conversion from Freebase to Wikidata to make sure that the answers are correct and relevant.
>
> To convert Freebase entities and properties to their Wikidata equivalents, our automatic conversion tool uses the mappings provided by Google and Wikidata. However, the mappings do not cover all the entities and properties. Additionally, the automatically converted SPARQL queries may not yield the expected results, as the information might be represented using a different property with similar semantics in Wikidata. Thus, the automatic tool can only convert 60% of the original dataset.
>
> It took approximately 4 weeks to complete manual annotation by a programmer working full time. To improve the reliability of manual conversion, we also developed a tool for the human annotator with automatic validation. This tool first checks the syntax of the SPARQL and then executes it to retrieve answers from Wikidata. The retrieved answers are shown to the annotator along with the original answer in the WebQuestions dataset. If the answers are different, the annotator will revisit the question to double-check if the SPARQL is correct.
>
> Despite being a much larger knowledge base, Wikidata is not a superset of Freebase. For example, flight information such as "which airport to fly into Rome?" is not available in Wikidata, but can be answered by Freebase.  We drop examples from WebQuestions if and only if the question cannot be answered by Wikidata.
>
> > Are constrained decoding techniques like PICARD (Scholak2021, https://github.com/ServiceNow/picard) considered for decoding the SPARQL queries?
>
> Thank you for suggesting this approach.
>
> From our error analysis, we did not observe any SPARQL syntax error in the model predictions. We did, however, encounter a few cases of imaginative properties. Unfortunately, with the Wikidata semantic graph, we do not know what properties are available for a specific entity, unlike databases with fixed schemas.
>
> > I am not sure how the backoff to GPT3 system would work? When is model not confident enough to backoff to GPT3?
>
> Thank you for this question. Indeed, we should have clarified how such a system works.
>
> We will add a diagram to illustrate the flow of our system to Fig. 1:
>
> ```
> Query → WikiSP (Semantic parser) → SPARQL → Wikidata → Answer    (1)
> if Answer is null, Query →  GPT-3 → Qualified answer             (2)
> else displays answer                                             (3)
>     if user downvotes answer, Query → GPT-3 → Qualified answer   (4)
> ```
> - (1) WikiSP parses user query to SPARQL, which is then used to fetch result from Wikidata;
>
> - (2) If Wikidata returns no result, this means that we may have gotten the parse wrong, or Wikidata has no answer, we use GPT-3 to return a qualified answer to the user;
>
> - (3) If there is a result, the result is displayed within the full context of the SPARQL query;
>
> - (4) The user can "downvote" our response, e.g. if there is a mismatch between the user’s intended query and the semantic parser’s prediction. In such cases, we follow up with GPT’s response, by saying that: "Here’s GPT-3’s unverified answer:  …"
>
> 1. Example with correct semantic parse
>
>     - User: "where was anne hathaway born?"
>
>     - Predicted SPARQL: `SELECT DISTINCT ?x WHERE { wd:Q36301 wdt:place_of_birth ?x. }`
>
>     - Our agent: "From Wikidata, the place of birth of Anne Hathaway is New York City."
>
> 2. Example with incorrect semantic parse
>
>     - User: "where did bronx tale take place?"
>
>     - Predicted SPARQL: `SELECT DISTINCT ?x WHERE { wd:Q1130705 wdt:filming_location ?x. }` *(where the correct SPARQL should use property narrative location instead of filming location)*
>
>     - Our agent: "From Wikidata, the filming location of ‘A Bronx Tale’ includes New Jersey and New York City."
>
>     - User: downvote this response *(since they notice that the agent is answering the wrong question)*.
>
>     - Our agent: "Here’s GPT-3’s unverified answer: The movie took place in the Bronx, New York."
>
> > Can you cite sources for "This may cause significant harm as people increasingly accept LLMs as a knowledge source." (line 38-39)
>
> Certainly! Here are some sources:
>
> [1] Benjamin Weiser. 2023. [Here’s What Happens When Your Lawyer Uses ChatGPT]( https://www.nytimes.com/2023/05/27/nyregion/avianca-airline-lawsuit-chatgpt.html ). *The New York Times.*
>
> [2] Bar Lanyado. 2023. [Can you trust ChatGPT’s package recommendations?]( https://vulcan.io/blog/ai-hallucinations-package-risk ). *vulcan.io*.
>
> [3] Jerome Goddard. 2023. [Hallucinations in ChatGPT: A Cautionary Tale for Biomedical Researchers]( https://www.amjmed.com/article/S0002-9343(23)00401-1/fulltext ). *Commentary in The American Journal of Medicine.*

---

### Official Review · Reviewer_1ArN · 2023-08-04

**Soundness:** 4

**Excitement:**

4: Strong: This paper deepens the understanding of some phenomenon or lowers the barriers to an existing research direction.

**Paper Topic And Main Contributions:**

Summary: For the task of knowledge base question answering (KBQA), the researchers study a means of increasing user confidence in an answer suggested by a LLM for a given question. The problem is that LLMs hallucinate answers, a problem that leads to users having to verify the factuality of all generated data. How could we get reassurance about an answer to a question? The first step is to generate a SPARQL query using a fine-tuned language model, retrieve the corresponding answer in a KB, and present that to the user alongside the LLM’s guess as to the answer to the original query.

To make a dataset to study this problem, researcher adapt an earlier version of web questions paired with executable queries originally made for Freebase. The researchers use this new dataset, WikiWebQuestions, for few-shot sequence-2-sequence SPARQL parsing using LLaMA. For encoding, the researchers test using a named entity disambiguator to identify entities first, appending entity mentions to the natural language query. For decoding, they test generating natural language strings for properties and entities in the generated SPARQL query rather than QIDs and PIDs, with the assumption that the LLM generations using natural language strings will be better formed (more likely executable). Results are reported with error analyses.


**Questions For The Authors:**

No examples or citations for claims made in first paragraph (this could be a footnote, “as reported in news media …”, or as part of a Figure 1).

(Line 39). What is this “alarm” based on? In which contexts do people “rely on LLMs as a knowledge source”? What are the high-stakes scenarios, for example?

(Lines 70-71; Line 78): Citations needed.

**Reasons To Accept:**

Strengths

- Novel means of fact-checking to improve trustworthiness in a LLM. Using semantic parsing as a step for factuality checking is a great idea and suggests hypotheses that are readily testable.

- Revamped dataset. Making a dataset of SPARQL queries has many challenges no matter its size, and the researchers here present one that is fairly large and appears challenging (although that is hard to determine without any examples given or other data available). The quality of the dataset (half automatically converted well with the remaining half being manually fixed) may be demonstrated in the results of the task, which suggest that this LLM can be fine-tuned to generate valid SPARQL queries for 72% of the examples in the dev set.

- An analysis that provides insight into SPARQL generation using LLMs by both considering the data, the prompts, and additional tools to add to the pipeline (entity linking).

**Reasons To Reject:**

Weaknesses:

- Not so helpful Figure 1. The figure shows the end results, but a more impactful Figure 1 would illustrate better more exactly what the problem is and what the plan is to address it.

- (Section 3.2.2) Recovering from NED errors. I had to reread this section a few times and still wasn’t sure of some things. Including examples in the Appendix would be very helpful.

- Missing information about the dev set.
  - Since this is the main data discussed in the paper, it would be very helpful to have some statistics about it.
  - Is the dev set very diverse in terms of types of questions, relations in answers, complexity of queries, and the number of answers that are possible? This would help better understand evaluation, for example how the researchers verified GPT-3 answers.
  - No examples of generated SPARQL compared with gold queries in the dev set.

- (Section 4) Missing details about how the dataset was made.
  - Consider discussing an evaluation of the automatic conversion of the dataset (section 4), through an analysis of the “60%” of the dataset that was automatically converted well (line 424). Also consider discussing the reliability of one person in converting the rest (which is about ~2000 examples) How long did that take? How could we verify the quality of that manual conversion?

  - (4. 1) Porting dataset to Wikidata from Freebase: How to decide if a question is not covered by Wikidata? How reliable is that procedure (lines 431-436)?

- (Section 6.2) The demonstration of existing tools as part of the pipeline, specifically the use an entity linker, appears well founded, but there is no evidence given to support its inclusion. Why didn’t the ablation study also include a step where the entity linking stage is completely removed? This could be the secret sauce, so I’d be interested in a comparison with input as just raw text.

- Missing future directions. What do your results suggest for the next iteration of the research cycle?

**Reproducibility:**

3: Could reproduce the results with some difficulty. The settings of parameters are underspecified or subjectively determined; the training/evaluation data are not widely available.

**Reviewer Confidence:**

4: Quite sure. I tried to check the important points carefully. It's unlikely, though conceivable, that I missed something that should affect my ratings.

---

> ### Author Rebuttal · Authors · 2023-08-29
>
> Thank you for your review and valuable comments.
>
> > Not so helpful Figure 1. The figure shows the end results, but a more impactful Figure 1 would illustrate better more exactly what the problem is and what the plan is to address it.
>
>
> Thank you for pointing this issue out.
>
> We will add a diagram to illustrate the user interaction flow to Fig. 1:
> ```
> Query → WikiSP (Semantic parser) → SPARQL → Wikidata → Answer
> if Answer is null, Query →  GPT-3 → qualified answer
> else displays answer
>     if user downvotes answer, Query → GPT-3 → qualified answer
> ```
>
> 1. Example with correct semantic parse
>
>     - User: "where was anne hathaway born?"
>
>     - Predicted SPARQL: `SELECT DISTINCT ?x WHERE { wd:Q36301 wdt:place_of_birth ?x. }`
>
>     - Our agent: "From Wikidata, the place of birth of Anne Hathaway is New York City."
>
> 2. Example with incorrect semantic parse
>
>     - User: "where did bronx tale take place?"
>
>     - Predicted SPARQL: `SELECT DISTINCT ?x WHERE { wd:Q1130705 wdt:filming_location ?x. }` *(where the correct SPARQL should use property narrative location instead of filming location)*
>
>     - Our agent: "From Wikidata, the filming location of ‘A Bronx Tale’ includes New Jersey and New York City."
>
>     - User: downvote this response *(since they notice that the agent is answering the wrong question)*.
>
>     - Our agent: "Here’s GPT-3’s unverified answer: The movie took place in the Bronx, New York."
>
> > (Section 3.2.2) Recovering from NED errors. I had to reread this section a few times and still wasn’t sure of some things. Including examples in the Appendix would be very helpful.
>
> Thank you for this valuable suggestion. We will include more concrete examples in the appendix. For instance, in the training set, we have the following example:
> - Query: "what year did giants win the world series?"
> - Gold SPARQL is:
> ```
> SELECT DISTINCT ?x WHERE {
>   ?y wdt:sports_season_of_league_or_competition wd:Q265538;
>   wdt:winner wd:Q308966;
>   wdt:point_in_time ?x. }
> ```
> - Gold NED result: "World Series with QID Q265538;
>                              San Francisco Giants with QID Q308966;"
> - Refined NED result: "San Francisco Giants with QID Q308966;"
>
> *(Here, the refined NED linker fails to identify the “World Series” entity.)*
>
> Our proposal of mentions gives the semantic parser a chance to recover the missing QID.
> To train the parser to generate mentions, our training includes samples like this:
>
> - Query: "what year did giants win the world series?"
>
> - NED result: "San Francisco Giants with QID Q308966;"
>
> - Gold target:
> ```
> SELECT DISTINCT ?x WHERE {
>   ?y wdt:sports_season_of_league_or_competition wd:world_series;
>   wdt:winner wd:Q308966;
>   wdt:point_in_time ?x. }
> ```
> The gold query mentions `world_series`. At inference time, our heuristics use the predicted mention to look up the actual Wikidata entity. For example, if `wd:world_series` is predicted at inference time, our heuristics maps it back to `Q265538`.
>
> > Missing information about the dev set. Since this is the main data discussed in the paper, it would be very helpful to have some statistics about it. Is the dev set very diverse in terms of types of questions, relations in answers, complexity of queries, and the number of answers that are possible? This would help better understand evaluation, for example how the researchers verified GPT-3 answers. No examples of generated SPARQL compared with gold queries in the dev set.
>
> Thank you for pointing this out. Our dataset was derived from WebQuestionSP, for which the statistics have been reported in prior work, e.g., Table 1 of [Gu et al. 2021](https://arxiv.org/abs/2011.07743). Certain natural language related statistics were also reported by the original authors of WebQuestion ([Berant et al. 2013]( https://aclanthology.org/D13-1160/ )) and WebQuestionSP ([Yih et al. 2016]( https://aclanthology.org/P16-2033/ )). Consequently, we share a similar set of characteristics, with minor differences originating from our conversion procedure (Section 4.1). We will make sure we cite these references.
>
> > (Section 4) Missing details about how the dataset was made. Consider discussing an evaluation of the automatic conversion of the dataset (section 4), through an analysis of the “60%” of the dataset that was automatically converted well (line 424). Also consider discussing the reliability of one person in converting the rest (which is about ~2000 examples) How long did that take? How could we verify the quality of that manual conversion? (4. 1) Porting dataset to Wikidata from Freebase: How to decide if a question is not covered by Wikidata? How reliable is that procedure (lines 431-436)?
>
> To convert Freebase entities and properties to their Wikidata equivalents, our automatic conversion tool uses the mappings provided by Google and Wikidata. However, the mappings do not cover all the entities and properties. Additionally, the automatically converted SPARQL queries may not yield the expected results, as the information might be represented using a different property with similar semantics in Wikidata. Thus, the automatic tool can only convert 60% of the original dataset.
>
> It took approximately 4 weeks to complete manual annotation by a programmer working full time. To improve the reliability of manual conversion, we also developed a tool for the human annotator with automatic validation. This tool first checks the syntax of the SPARQL, and then executes it to retrieve answers from Wikidata. The retrieved answers are shown to the annotator along with the original answer in the WebQuestions dataset. If the answers are different, the annotator will revisit the question to double-check if the SPARQL is correct.
>
> Despite being a much larger knowledge base, Wikidata is not a superset of Freebase. For example, flight information such as "which airport to fly into Rome?" is not available in Wikidata, but can be answered by Freebase. We drop examples from WebQuestions if and only if the question cannot be answered by Wikidata.
>
> > (Section 6.2) The demonstration of existing tools as part of the pipeline, specifically the use an entity linker, appears well founded, but there is no evidence given to support its inclusion. Why didn’t the ablation study also include a step where the entity linking stage is completely removed? This could be the secret sauce, so I’d be interested in a comparison with input as just raw text.
>
> Thank you for this valuable suggestion, we will carry out an ablation experiment in the final paper.
>
> Many related semantic parsing over KBQA tasks use an entity linker step: [Gu et al. 2021]( https://arxiv.org/abs/2011.07743 ) (the need for an entity linker for a large-scale KBQA system is discussed under Section 3.2) and [Chen et al. 2021]( https://aclanthology.org/2021.acl-demo.39/ ).
>
> In our dev set, there are many cases where translating entity mentions to their QIDs is challenging. Common examples include acronyms (e.g. "what does time warner own?", where time warner refers to WarnerMedia) and contextual omissions (e.g. "who was darth vader in episode 3?", where episode 3 refers to "Star Wars: Episode III – Revenge of the Sith", "who are the two state senators of georgia?" where georgia refers to the state). These examples illustrate the importance of a neural NED. Generally, the need for an entity linker stage is also needed when dealing with tail entities and a customized knowledge base.
>
> > Missing future directions. What do your results suggest for the next iteration of the research cycle?
>
> Thank you for the opportunity to discuss the next iteration of our research! Our future direction is based on an additional experiment we performed for Reviewer 3 on the QALD-7 ([Usbeck et al. 2017]( https://svn.aksw.org/papers/2017/ESWC_2017_QALD/public.pdf )) dataset. We will first discuss the results of the experiment to demonstrate how our findings motivated future research.
>
> **Why we choose QALD7 and not other datasets**: In this paper, we focus on question answering on Wikidata and we avoid datasets built on synthetic data or human paraphrase data, such as CSQA ([Saha et al. 2018]( https://ojs.aaai.org/index.php/AAAI/article/view/11332 ))  and KQA-Pro ([Cao et al. 2022]( https://aclanthology.org/2022.acl-long.422/ )). For synthetic or human paraphrase data, due to the extremely limited natural language variety shared between the training and evaluation data, models can get artificially high accuracy. For example, a simple BART model can achieve over 90% accuracy on KQA-Pro even without an entity linking module.
>
> Thus to add another dataset as per request of Reviewer 3 we chose QALD-7. This dataset is part of the QALD (Question Answering over Linked Data) challenges, a popular series of challenges started in 2011 known for their complex, manually created questions. It mainly focuses on DBpedia, but QALD-7 's Task 4 is engineered for Wikidata. The task includes 100 train examples, which we use to finetune our model, and 50 test examples for evaluations.
>
> **Experimental results**: The QALD-7 test set contains both SPARQL queries as well as the answers. To double-check the correctness of the QALD-7 dataset, we applied the 50 gold queries of the test set to Wikidata and found that 4 did not return an answer. We hypothesize that the discrepancy is caused by the change in Wikidata structure/quantity of information. Thus the results below are reported only on the 46 remaining. Note that the prior work we are comparing to evaluates on all 50.
>
> For this experiment, we use the same hyperparameters and data format as described in Section 5.2. In addition to the training data for WikiSP, we also include the QALD-7 train examples, upsampled by 20 times.
>
> QALD-7 ([Usbeck et al. 2017]( https://svn.aksw.org/papers/2017/ESWC_2017_QALD/public.pdf )) reports that the leader WDAqua gets an F1 score of 0.552, but the paper of WDAqua  ([Diefenbach et al. 2017]( https://link.springer.com/chapter/10.1007/978-3-319-69146-6_8 )) reports 0.40. The runner-up reports an F1 score of 0.427. Our model achieves an F1 score of 0.47. Note that WDAqua is based on retrieval, whereas WIkiSP is based on semantic parsing. (The differences between retrieval and semantic parsing are discussed in Section 2.1)
>
> One of the authors who was not involved in training or fine-tuning the model evaluated GPT-3 on the test set. GPT-3 is fully accurate on 63% of the questions, 20% incomplete, 16% wrong. WikiSP gets 41% of the questions correct.  So we can provide 41% verifiably good answers; the guesses of GPT-3 get an additional 33% correct, 15% incomplete, and 11% wrong. It was noted that the test set is significantly more complicated than the training data (of just 100 samples), with most of the queries containing multiple properties, thus explaining the lower accuracy. (We do not report a detailed error analysis because there is no dev set).
>
> **Future direction**: The results suggest that an important future direction is to improve the few-shot training data by including more complex queries.
>
> > No examples or citations for claims made in first paragraph (this could be a footnote, “as reported in news media …”, or as part of a Figure 1). (Line 39). What is this “alarm” based on? In which contexts do people “rely on LLMs as a knowledge source”? What are the high-stakes scenarios, for example? (Lines 70-71; Line 78): Citations needed.
>
> Thank you for pointing these issues out! Here are some reports and citations for the high-risk scenarios discussed in paragraph 1:
>
> - [Weiser. 2023]( https://www.nytimes.com/2023/05/27/nyregion/avianca-airline-lawsuit-chatgpt.html )
> - [Lanyado. 2023]( https://vulcan.io/blog/ai-hallucinations-package-risk )
> - [Goddard. 2023]( https://www.amjmed.com/article/S0002-9343(23)00401-1/fulltext )
>
> Here are citations for lines 70-71, "Suffering from limited training data compared to the massive knowledgebase, seq-2-seq approach is outperformed by approaches using subgraph extraction":
>
> - [Yin et al. 2021]( https://www.sciencedirect.com/science/article/pii/S0167739X20330752?via%3Dihub )
> - [Gu et al. 2021]( https://arxiv.org/abs/2011.07743 )
> - [Banerjee et al. 2022]( https://dl.acm.org/doi/10.1145/3477495.3531841 )
>
> Citation for Line 78, "Pretrained with internet corpora, LLMs are already familiar with formal query languages such as SQL ([Li et al. 2023]( https://arxiv.org/abs/2305.03111 )) and SPARQL ([Lubiana. 2023]( https://www.wisecube.ai/blog/sparql-queries-gpts-and-large-language-models-where-are-we-currently/ ))."
>
> ### References
>
> [1] Yu Gu, Sue Kase, Michelle Vanni, Brian Sadler, Percy Liang, Xifeng Yan, and Yu Su. 2021. [Beyond I.I.D.: Three Levels of Generalization for Question Answering on Knowledge Bases]( https://arxiv.org/abs/2011.07743 ). *Proceedings of the Web Conference 2021.*
>
> [2] Jonathan Berant, Andrew Chou, Roy Frostig, and Percy Liang. 2013. [Semantic Parsing on Freebase from Question-Answer Pairs]( https://aclanthology.org/D13-1160/ ). *Proceedings of the 2013 Conference on Empirical Methods in Natural Language Processing.*
>
> [3] Wen-tau Yih, Matthew Richardson, Chris Meek, Ming-Wei Chang, and Jina Suh. 2016. [The Value of Semantic Parse Labeling for Knowledge Base Question Answering]( https://aclanthology.org/P16-2033/ ). *Proceedings of the 54th Annual Meeting of the Association for Computational Linguistics (Volume 2: Short Papers). *
>
> [4] Shuang Chen, Qian Liu, Zhiwei Yu, Chin-Yew Lin, Jian-Guang Lou, and Feng Jiang. 2021. [ReTraCk: A Flexible and Efficient Framework for Knowledge Base Question Answering]( https://aclanthology.org/2021.acl-demo.39/ ). *Proceedings of the 59th Annual Meeting of the Association for Computational Linguistics and the 11th International Joint Conference on Natural Language Processing: System Demonstrations*
>
> [5] Ricardo Usbeck, Axel-Cyrille Ngonga Ngomo, Bastian Haarmann, Anastasia Krithara, Michael Röder, and Giulio Napolitano. 2017. [7th Open Challenge on Question Answering over LinkedData (QALD-7)]( https://svn.aksw.org/papers/2017/ESWC_2017_QALD/public.pdf ).
>
> [6] Amrita Saha, Vardaan Pahuja, Mitesh Khapra, Karthik Sankaranarayanan, and Sarath Chandar. 2018. [Complex Sequential Question Answering: Towards Learning to Converse Over Linked Question Answer Pairs with a Knowledge Graph]( https://ojs.aaai.org/index.php/AAAI/article/view/11332 ). *Proceedings of the AAAI Conference on Artificial Intelligence, 2018. *
>
> [7] Shulin Cao, Jiaxin Shi, Liangming Pan, Lunyiu Nie, Yutong Xiang, Lei Hou, Juanzi Li, Bin He, and Hanwang Zhang. 2022. [KQA Pro: A Dataset with Explicit Compositional Programs for Complex Question Answering over Knowledge Base]( https://aclanthology.org/2022.acl-long.422/ ). *Proceedings of the 60th Annual Meeting of the Association for Computational Linguistics (Volume 1: Long Papers).*
>
> [8] Dennis Diefenbach, Kamal Singh, and Pierre Maret. 2017. [WDAqua-core0: A Question Answering Component for the Research Community]( https://link.springer.com/chapter/10.1007/978-3-319-69146-6_8 ). *SemWebEval 2017. Communications in Computer and Information Science, vol 769.*
>
> [9] Benjamin Weiser. 2023. [Here’s What Happens When Your Lawyer Uses ChatGPT]( https://www.nytimes.com/2023/05/27/nyregion/avianca-airline-lawsuit-chatgpt.html ). *The New York Times.*
>
> [10] Bar Lanyado. 2023. [Can you trust ChatGPT’s package recommendations?]( https://vulcan.io/blog/ai-hallucinations-package-risk ). *vulcan.io.*
>
> [11] Jerome Goddard. 2023. [Hallucinations in ChatGPT: A Cautionary Tale for Biomedical Researchers](https://www.amjmed.com/article/S0002-9343(23)00401-1/fulltext). *Commentary in The American Journal of Medicine.*
>
> [12] Xiaoyu Yin, Dagmar Gromann, and Sebastian Rudolph. 2021. [Neural machine translating from natural language to sparql](https://www.sciencedirect.com/science/article/pii/S0167739X20330752?via%3Dihub). *Future Generation Computer Systems, 117:510–519.*
>
> [13] Debayan Banerjee, Pranav Ajit Nair, Jivat Neet Kaur, Ricardo Usbeck, and Chris Biemann. 2022. [Modern baselines for sparql semantic parsing.](https://dl.acm.org/doi/10.1145/3477495.3531841). *In Proceedings of the 45th International ACM SIGIR Conference on Research and Development in Information Retrieval, SIGIR ’22, page 2260–2265, New York, NY, USA. Association for Computing Machinery.*
>
> [14] Jinyang Li, Binyuan Hui, Ge Qu, Binhua Li, Jiaxi Yang, Bowen Li, Bailin Wang, Bowen Qin, Rongyu Cao, Ruiying Geng, Nan Huo, Xuanhe Zhou, Chenhao Ma, Guoliang Li, Kevin C.C. Chang, Fei Huang, Reynold Cheng, and Yongbin Li. 2023. [Can LLM Already Serve as A Database Interface? A BIg Bench for Large-Scale Database Grounded Text-to-SQLs](https://arxiv.org/abs/2305.03111).
>
> [15] Tiago Lubiana. 2023. [Sparql Queries, Gpts and Large Language Models – Where Are We Currently?]( www.wisecube.ai/blog/sparql-queries-gpts-and-large-language-models-where-are-we-currently ) *Wisecube AI*.

---

### Meta-Review · Area_Chair_byRH · 2023-09-19

**Recommendation:** 4

**Metareview:**

Paper Topic And Main Contributions:
* Introduction of a novel semantic parser for Wikidata, accompanied by a new dataset called WikiWebQuestions. This dataset contains semantic parses over Wikidata, providing a valuable resource for training and evaluating semantic parsers in the context of Wikidata.
* The proposed semantic parser, WikiSP, is developed by fine-tuning LLama with Alpaca instruction tuning set and modified SPARQL queries.
* Extensive evaluation on the proposed WikiWebQuestions dataset offers insights into the performance of the semantic parser. By highlighting different categories of errors, the evaluation provides a roadmap for future research and the development of improved techniques in semantic parsing for Wikidata.
* In addition to WikiSP, the paper introduces a shallow integration of WikiSP with GPT3. The semantic parser initially responds to user queries, and its outputs are subsequently supplemented and backed up with responses from GPT3, providing a two-step approach to generate more comprehensive and accurate answers.

Reasons to accept:
* Introduction of a new dataset, WikiWeb-Questions, which is a high-quality knowledge base question answering benchmark for Wikidata.
* The paper presents the first effective few-shot sequence-to-sequence semantic parser for Wikidata.
* The proposed methodology reduces the hallucination of large language models like GPT-3 by grounding it with a semantic parser for Wikidata.

Reasons to reject:
* This paper only conducts experiments on a single dataset (self-collected), but other KBQA datasets exist (e.g., WebQuestions, GrailQA). The authors address this in the rebuttal by adding experiments on QALD-7. They also clarify that they focus on Wikidata.

---

### Decision · Program_Chairs · 2023-10-07

**Decision:**

Accept-Main

**Comment:**

Paper Topic And Main Contributions:
* Introduction of a novel semantic parser for Wikidata, accompanied by a new dataset called WikiWebQuestions. This dataset contains semantic parses over Wikidata, providing a valuable resource for training and evaluating semantic parsers in the context of Wikidata.
* The proposed semantic parser, WikiSP, is developed by fine-tuning LLama with Alpaca instruction tuning set and modified SPARQL queries.
* Extensive evaluation on the proposed WikiWebQuestions dataset offers insights into the performance of the semantic parser. By highlighting different categories of errors, the evaluation provides a roadmap for future research and the development of improved techniques in semantic parsing for Wikidata.
* In addition to WikiSP, the paper introduces a shallow integration of WikiSP with GPT3. The semantic parser initially responds to user queries, and its outputs are subsequently supplemented and backed up with responses from GPT3, providing a two-step approach to generate more comprehensive and accurate answers.

Reasons to accept:
* Introduction of a new dataset, WikiWeb-Questions, which is a high-quality knowledge base question answering benchmark for Wikidata.
* The paper presents the first effective few-shot sequence-to-sequence semantic parser for Wikidata.
* The proposed methodology reduces the hallucination of large language models like GPT-3 by grounding it with a semantic parser for Wikidata.

Reasons to reject:
* This paper only conducts experiments on a single dataset (self-collected), but other KBQA datasets exist (e.g., WebQuestions, GrailQA). The authors address this in the rebuttal by adding experiments on QALD-7. They also clarify that they focus on Wikidata.